# Characterization and Evaluation of Human–Exoskeleton Interaction Dynamics: A Review

**DOI:** 10.3390/s22113993

**Published:** 2022-05-25

**Authors:** Stefano Massardi, David Rodriguez-Cianca, David Pinto-Fernandez, Juan C. Moreno, Matteo Lancini, Diego Torricelli

**Affiliations:** 1Neural Rehabilitation Group, Cajal Institute, Spanish National Research Council (CSIC), 28006 Madrid, Spain; stefano.massardi@cajal.csic.es (S.M.); david.rodriguez.cianca@csic.es (D.R.-C.); david.pinto@cajal.csic.es (D.P.-F.); jc.moreno@csic.es (J.C.M.); 2Department of Mechanical and Industrial Engineering (DIMI), University of Brescia, 25100 Brescia, Italy; 3Universidad Politécnica de Madrid (UPM), 28040 Madrid, Spain; 4Department of Medical and Surgical Specialties, Radiological Sciences and Public Health (DSMC), University of Brescia, 25100 Brescia, Italy; matteo.lancini@unibs.it

**Keywords:** exoskeletons, wearable robots, physical human–exoskeleton interaction, safety, forces, pressures

## Abstract

Exoskeletons and exosuits have witnessed unprecedented growth in recent years, especially in the medical and industrial sectors. In order to be successfully integrated into the current society, these devices must comply with several commercialization rules and safety standards. Due to their intrinsic coupling with human limbs, one of the main challenges is to test and prove the quality of physical interaction with humans. However, the study of physical human–exoskeleton interactions (pHEI) has been poorly addressed in the literature. Understanding and identifying the technological ways to assess pHEI is necessary for the future acceptance and large-scale use of these devices. The harmonization of these evaluation processes represents a key factor in building a still missing accepted framework to inform human–device contact safety. In this review, we identify, analyze, and discuss the metrics, testing procedures, and measurement devices used to assess pHEI in the last ten years. Furthermore, we discuss the role of pHEI in safety contact evaluation. We found a very heterogeneous panorama in terms of sensors and testing methods, which are still far from considering realistic conditions and use-cases. We identified the main gaps and drawbacks of current approaches, pointing towards a number of promising research directions. This review aspires to help the wearable robotics community find agreements on interaction quality and safety assessment testing procedures.

## 1. Introduction

Exoskeletons are starting to be extensively used in many applications, spanning from military to industrial use, personal care, and medical applications. This is reflected by an increasing trend in the number of devices present on the market [1]. Their range of applicability is expanding together with the evolution of automatized industrial processes—which still require the involvement of human workers [2,3]—and the aging of the population. Aging is associated with increasing mobility impairments, making the demand for rehabilitation and assistive devices grow every year [4]. The use of exoskeletons in rehabilitation medicines represents one of the most grounded scenarios in their future development. Applications such as supporting mobility of spinal cord injured (SCI) persons and rehabilitation of major trauma patients remains the primary focus of exoskeleton research [4].

Due to the increasingly aging population [5], new scenarios are starting to receive attention, not only in the field of after-treatment therapies but also to help elderly people to remain independent by providing daily life assistance [6]. In several of these activities, exoskeletons help users to perform tasks by providing assistance and augmentation of individual capabilities through increasing the range of motion of individual joints [7].

Safety and user acceptability will be the underlying evaluation criteria for the mechanical design, actuation, and control architectures of future exoskeleton developments [5]. One fundamental aspect that differentiates exoskeletons from other robotic technologies is the intrinsic and close physical interaction with humans, defined as the generation and exchange of a net flux of power between both actors [5]. ISO 13482:2014 [6] states how physical interaction (e.g., contact forces) between robot and human will be designed to be as low as reasonably practicable. However, current international standards do not provide realistic protocols to assess contact safety, and in the case of exoskeletons, regulatory gaps are yet to be addressed [7,8].

In exoskeletons, force or torque are usually transferred through attachment devices (i.e., connection cuffs or orthoses) [9] producing interaction forces that are key in the generation of shear stresses, interface movements, and misalignments [10]. This physical interaction between the human user and the wearable robot should be carefully monitored and controlled since an unexpected behavior by one of the actors during the task might have an impact on safety and the system. For this reason, a crucial challenge in exoskeleton design is to minimize the risks introduced by the dynamic interaction between the human and the exoskeleton’s physical interfaces. A truly ergonomic physical interface should be customized to an individual’s own anthropometrics and needs [11]. However, this concept is unlikely to be applicable to most current devices as they follow a design adaptation concept to fit a large spectrum of possible users. However, how the predefined interface can influence physical human–exoskeleton interaction (pHEI) remains to be defined in a more generalized way. There is still considerable room for the research on pHEI to grow compared with other more accepted fields. Current exoskeleton evaluation processes often include tests with humans for the evaluation of physiological, kinematic, and kinetic effects of human–device interaction [12], but metrics and protocols able to characterize pHEI are still not clarified, preventing standardized pHEI evaluations. Additionally, different metrics can be difficult to apply to a broad spectrum of devices, and the selection of an accepted relevant set of metrics is far from being accomplished. Testbed platforms for proving exoskeleton compliance with contact safety requirements are still limited and developed for specific device solutions [13], while traditional comfort evaluations are based on subjective pain rating scales [14,15,16]. Qualitative feedback can be improved by user-centered designs and individual needs assessments [17] but do not normally include quantitative measurements able to produce a well-accepted body of scientific knowledge in the field of pHEI. In this growing and partially unexplored field, the necessity to concentrate the future efforts in a common direction is a pressing requirement for the wearable robot community.

The aim of this work is to systematically review recent studies including pHEI measurements such as contact forces, torques, and pressures in order to summarize and analyze the current knowledge, techniques, and metrics used in pHEI measurement. This work answers to the need for building a comprehensive revision of pHEI-related works in an effort to create a first step in the acceptance of a shared set of metrics and methods in pHEI measurement and their safety assessment.

## 2. Materials and Methods

We define pHEI measurement as any extraction of information related to forces (including torques and pressures) exchanged between a human and an exoskeleton during the execution of a task. Considering the recent and rapid growth of exoskeletons in the last year, we decided not to include studies older than 10 years as they are likely considering depreciated and early-stage devices that are now better equipped and developed. Similar growth is also seen in the world of sensor technologies; for these reasons, various searches were conducted on the Scopus scientific database between 1 January 2010 and 31 December 2021. We looked for articles that included references to human–exoskeleton interaction databases using the AND/OR/NOT Boolean operators with different combinations of terms from 3 sets of keywords:exoskeleton*, physical assist*, wearable rob*;physical human-rob*, human-robot inter*, phri, pressure, safety;measure*, asses*, benchmar*, eval*.

The searches provided a list of 785 publications. After removing duplicated publications and a preliminary review of titles and abstracts, 121 publications were selected for a full text review. A total of 54 publications have been included in this work. The review’s flow diagram is depicted in Figure 1.

Kinematic or physiologically based metrics such as relative motions, discomfort, and fatigue do not provide direct information of pHEI but rather its consequences on the human body. For this reason, additional kinematic and physiological metrics are included in the results only when supported by pHEI measurements. Other measurements such as Ground reaction forces (GRFs) or Muscle activation (EMG) fell outside the research.

We decided to include both upper limb and lower limb exoskeletons since the proposed solutions can often be shared between the two. For the same reason, both powered exoskeleton, passive exoskeletons, and exosuits have been considered since interaction issues are common among wearable devices. Wrist and hand exoskeletons were excluded from this review since metrics, functions, and evaluations sensitively differ from lower and upper limb exoskeletons.

We classified the papers based on the metric extracted and the sensor solution adopted. The following definitions apply in this paper for pHEI measurement:Interaction forces: forces exchanged between the human body and wearable device.Interaction torques: torques produced by interaction forces.Interaction pressures: pressure calculated from interaction forces over a contact area.

pHEI metrics are classified as follows:Force metrics: metrics extracted from interaction force measurement, including normal and shear forces as well as overall interaction force, peak, and average contact force.Torque metrics: metrics extracted from interaction torque measurement, normally represented by the single interaction torque generated during the task.Pressure metrics: metrics extracted from interaction pressure measurement such as maximum pressure and pressure distribution.

Indirect pHEI metrics were also considered:Relative motions: relative motion (in one or more dimensions) between a defined part of the human body and the worn device (frame shift, skin slippage).Misalignment: Mismatch in the correspondence in position and orientation between the anatomical and device joint axes.Subjective experience metrics (SE): metrics extracted by means of live feedback or questionnaires (Table 1).

## 3. Results

Of the 54 publications selected, 33 (61%) were published in the last 5 years, from 2016 to 2021 (Figure 2).

Figure 3 shows the number of publications including each of the considered family of metrics divided into upper and lower limb studies. Interaction was mostly assessed at the lower limbs, with 34 publications (63%) using lower limb devices (including hip exoskeletons and passive leg orthoses) in comparison to the 23 results (42%) obtained for upper body exoskeletons (including shoulder, elbow, and arm support devices). Both groups are counting 3 publications including both upper and lower limb contact measurements. Force-related metrics were preponderant, with 42 publications (78%). Pressure and torque metrics were included in 19 (35%) and 16 (29%) results, respectively. We found a minor part of the results including supporting pHEI metrics such as user experience (15%), joint misalignments (9%), and relative motions (7%).

Metrics from Figure 3 are further detailed, dividing force metrics into overall force metrics, normal (perpendicular to the surface), tangential, and distributed force metrics. The same division is applied for pressure metrics excluding overall pressure since no metrics could fit. Results including these metrics are matched with the relative instrumentation used for their extraction. Figure 4 presents the number of results for each proposed metric, sensor solution, and the intersection between the two axes.

Concerning the instrumentation, load cells (1-axis, 3-axis, and 6-axis) were used in half of the works (26 publications, 48%) to extract force and torque metrics. Optical systems such as fiber optics and laser sensors accounted for 15% of the results. The use of sensors based on force sensing resistors (FSRs) was found in 13 publications (24%), while optical motion tracking systems and air-based pressure sensors (air cushions, pneumatic pads) were found in 13% of the results. Pressure pads different from the above-mentioned technologies were found in 4 studies (7%). The remaining solutions, i.e., strain gauges, goniometers, inclinometers, and capacitive sensors, were found in 16% of the results. Questionnaires were used to extract user-experience metrics and were only considered when pHEI measurements were also included, accounting for 6 publications (11% of the results). Other than questionnaires, visual analog scales (VAS) were also used to extract perceived discomfort [18]. Among the sensor solutions presented in the recordings, 14 (28%) proposed customized sensor solutions for their pHEI measurements [9,18,19,20,21,22,23,24,25,26,27,28,29,30,31,32,33,34,35,36]. Most of the developed solutions were FSR-based [19,25,30,33,35] or air-based pressure sensors [18,23,28,29,31]. Optical solutions were divided into optical-fiber sensors [26,27,36] and optoelectronic laser-based sensors [9,20,21]. The remaining solutions were composed of force sensor [22], tactile sensor [24], 3D-printed capacitive sensors [32], and elastic band able to measure interaction through its deformation [34].

Force metrics were preponderant, and specifically one-dimensional forces (typically normal to the contact) were generally used for control purposes [19,28,33,37,38,39,40,41] or for contact evaluation strategies such as interface design evaluation [35,42], human–device kinematic compatibility [43,44], misalignment evaluation [31,45,46,47,48], or intention detection [40,49]. Normal force metrics were generally the mean absolute value of normal force during the task [18,45,50], force root mean square (RMS) [51,52,53], average force in a cyclic task [9,39,53,54,55,56,57], peak force [9,58,59,60], and force range [61]. Normal force mapping allowed the detection of possible areas for interface improvements [24,30,35,42,52].

Normal forces were also used for pressure prediction in new sensory solutions [29]. Torque metrics are normally taken at the joint level to compute interaction torque transferred to the user through the physical interface. Torques can be used in pHEI models to compute how loads are transferred, used for device control [34,38,51,58,62,63], and pHEI prediction and estimation [29,45,46,52,53,64,65]. Pressure measurements were often accompanied by pressure distribution evaluations [20,21,23,25,32,66,67], followed by maximum pressure reached during the task [9,31,55,67,68]. Maximum shear pressure was found only in [56,61], while strapping pressure was also included in a minor part of the results [18,45].

All the studies including pHEI modelling also physically measured pHEI in accordance with our review requirements. The use of models for pHEI evaluation is still limited, with direct measurements being the preferred option. Seventeen results (20%) implemented contact modelization, but 10 of them were for control purposes [37,38,40,41,49,50,51,54,58,62,63], while 7 results modelled human–device interactions for pHEI prediction or evaluation [18,45,46,52,53,64,65]. Interaction was simply modelled with kinematic parameters for misalignment prediction [46]. Later, human–interface contact was modelled by a spring [45,52] or spring-damper element [18,58,65]. In [64], a more advanced model including the knee angle was needed because the spring damper was found to be insufficient to describe the interaction forces. Nonlinear spring-damper elements were suggested to better describe the contact behavior, at the cost of a higher associated uncertainty [65]. The stiffness and shape of the subject were claimed to change with motion. Therefore, an improved spring-damper-attitude model including limb position was needed for pHEI modelization in [53].

Twenty-two results (44%) focused on evaluating or improving pHEI safety [18,20,23,24,25,30,32,42,45,46,50,53,55,56,59,61,64,66,67,68,69,70]. However, 10 of them effectively compared results with safety references [18,20,25,45,55,56,59,66,68,69].

A minor part of these results considered shear pressures [56,61,69], whereas only two studies evaluated and applied safety thresholds [56,69].

Extensive presentation of the results is shown in Table 2, listing results by first author, year of publication, metrics, sensors for their extraction, synthetized protocol applied, device used, and sensorized part of the body.

**Table 1 sensors-22-03993-t001:** Results including questionnaires with the related extracted metrics.

Ref.	Questionnaire	Output
[45]	NASA TLX [71]	Comfort, Physical demand, Mental demand, Temporal demand, operator performance, Effort
[47]	Custom Borg scale [72]	Perceived comfort, Physical load
[54]	Custom	Comfort, interface preference
[55]	Custom	Comfort
[70]	Custom	Safety
[68]	Borg category ratio (CR-10) [72] Van der Grinten and Smitt System Usability Scale (SUS)	Perceived musculoskeletal effort (arm, trunk, leg) Local Perceived Pressure (back/shoulders, arms, chest, and belly/hips) Usability of the exoskeleton

## 4. Discussion

Research in pHEI has been growing in the last years together with the necessity to properly address safety issues in wearable robots. However, the gap between the current knowledge and the need for data-based evidence to test and prove the level of safety in the current growing market is still huge. So far, the great variability of pHEI assessment methods, devices, and applications has prevented their harmonization. The need for a more standardized way to evaluate the safety of human–exoskeleton contact issues is becoming urgent. In this section, we summarize the main metrics found for pHEI measurement, discussing their implementation and the challenges in building links between measurements and safety.

### 4.1. pHEI Metrics and Measurements

Force was the principal quantity extracted to address pHEI, typically assessed in a normal direction with the body (normal interaction force).

Maximum normal force is typically used to assess and evaluate new solutions by monitoring their ability to decrease force magnitude. However, if the residual interaction is safe for the user, it is usually not addressed [44,59]. Tangential, or shear, forces can be extracted together with the normal component adding important information about the contact, being responsible for twisting and tearing the user’s skin, which can produce discomfort and skin injuries [73]. Metrics including shear forces can be monitored to inform interaction quality and power transfer efficiency, although some sensor solutions cannot properly address their contribution in the overall interaction. FSR-based solutions were never characterized with tangential interaction while other solutions simply recorded the overall interaction force (pneumatic/air-based sensors). Shear contribution was generally rarely analyzed.

Force and torque metrics were also included in interaction models for control and pHEI prediction. Human body parameters can drastically change among subjects and body sites [18]. Results are affected by individuality [56], making model robustness one of the major challenges in the field. Model complexity and limited reproducibility of exoskeleton tasks prevents them from reaching accurate predictions outside the experimented scenario, thus limiting the findings to the single tests [32,64] and leading to a preference for simpler models [52,65]. However, we consider research on pHEI modelization to be important since the pHEI sensory system is often applicable for specific conditions, and new knowledge from modelization would allow pHEI predictions and monitoring through a set of standard sensors outside the testing phase.

From this perspective, only a limited portion of the results considered pressure measurements and related metrics. Difficult contact area estimation and expensive solutions still prevent the community from attaining comprehensive knowledge of interaction pressure. Pressure measurements were often related to the study of distribution and how they vary with time and motion. How this pressure changes in time and space during the motion, however, is still not fully addressed. Dividing tangential from perpendicular interaction is generally difficult, and the real contribution of each component is often unknown [31]. The lack of evidence of human limits under shear stress together with the technological challenges resulted in a very restricted number of publications including shear pressures [56,61]. Both results extracted shear pressure by dividing tangential forces over an estimated contact area. At the current state of the art, no solution was found that could provide shear pressure output. This result shows how research in this field is influenced by technological limitations. Independently from the growth and interest in the field, we consider that facing the complex contact behaviour at the interface represents one of the main factors hampering new findings in pHEI. Calibration with shear forces is normally difficult and performed on flat and rigid surfaces. Interaction on soft human tissues introduces serious challenges in terms of test reproducibility sensor positioning and output calibration that hamper the development of a clear setup for pHEI evaluation. Any new attempt in this direction can produce precious and unexplored knowledge of how human–exoskeleton contact is characterized, and what the implications are during contact safety evaluation. At the time of this review, efforts in this direction are still limited.

Other metrics can inform the quality and safety level of the interaction, but their meaning needs to be clarified through pHEI measurements (e.g., forces, torques, pressures) since they do not provide direct information on the physical contact. Relative motions can describe how efficiently power can be transmitted from the device to the user’s biological structures [74]. This information is usually extracted through optical systems. A few exceptions are represented by laser sensors monitoring skin slippage and velocity [64,69].

Other kinematic metrics, such as human–exoskeleton joint misalignments, are responsible for undesired shear forces at the interface. Misalignments represent an important field of study. They can achieve the order of ±10 cm in various directions, even if at the start of the movement joints are well aligned [45]. Misalignments cause frame shifts and limit the voluntary range of motion, especially in larger motions, together with the generation of undesired forces. However, a clear linear dependence between misalignments and force is not always observed, suggesting a more complex relationship. A proper misalignment definition will be evaluated by checking both the 3-D position and angular alignment between the joint centres, although they are often reduced to a single plane. At the state of the art, no results were found considering joint misalignments in the 3D space. Defining the degree of relevance of the different planes/directions will be an issue for future studies. The same considerations apply to relative motions and slippage since they are always measured taking into account fewer dimensions and plans.

We consider that the relationship between kinematic metrics and forces will be matters of further investigation in future studies The possibility of informing some safety aspects by means of kinematic measurements may represent a promising approach in the field of pHEI, preventing the experimenters from dealing with the difficulties of force/pressure measurements.

Nevertheless, strong preliminary studies are first needed to support pHEI evaluations through kinematic metrics, and few efforts were found in this direction.

User experience assessment is normally conducted using questionnaires and open feedback from users. Typical metrics were perceived pressure, comfort, and safety during the task. Their assessment, together with pHEI measurements, can provide useful links between subjective and physical metrics in the effort to find physiological limits associated with safety. We could not find any publication linking physical metrics with perceived pain or discomfort during an exoskeleton task. User experience metrics were mainly used as feedback for the assessment of the device or the additional mechanism applied.

With regard to the sensing systems, load cells were often included for force measuring. Depending on the available degrees of freedom, load cells can extract different metrics such as normal interaction force (using 1-3-6 axis load cell), shear forces (3–6 axis load cell), and interaction moments (6 axis load cell). Load cells do not require particular models but do not allow a comprehensive quantification of how pHEI is distributed at the contact area, providing a limited representation of the overall behaviour. Load cells were normally integrated in the exoskeleton itself, thereby reducing the setup flexibility and applicability to other devices. An alternative solution to overcome the issue of force distribution was found in FSR-based sensors. Their main advantages are the reduced cost and the possibility to be comfortably placed at the contact points without affecting the user comfort. However, these solutions often suffer from drift caused by prolonged pressure and remain less suitable for bending and tearing. They need to be manually calibrated after placement and can only estimate normal forces. On the other side, thanks to their limited and known area, they have been used to map pressure distribution at the interfaces. Nevertheless, they can also suffer from poor repeatability, thus raising some concerns regarding their extensive use in pHEI applications.

Commercial sensors based on FSR technology could guarantee higher performance compared with customized solutions. Additionally, commercial solutions can often offer a more comfortable hardware wearing, limiting wiring and electronic devices. Their main drawback is their generally higher cost and more complex integrability in a wider experimental setup. Furthermore, attention will be paid to evaluating the comfort–pressure relation when adding pressure mats as they introduce additional stiffness that can sensitively change the perceived comfort between limb and interface [66]. An alternative and promising solution was represented by air-based sensors, such as pneumatic padding and air cushions [18]. These solutions can be inflated at the desired pressure, thus monitoring their compliance and providing absolute pressure value independently of the load direction. In this way, not only normal interaction pressure but the overall interaction can be recorded without the need for complicated calibration procedures. All air-based solutions in the literature were developed and built by the experimenters for the published applications. No commercial solutions were applied in the results.

A discrete number of publications developed customized sensor solutions, which highlights the lack of commercially available solutions able to meet the requirements and needs of pHEI assessment. Most of the solutions were based on FSR and pneumatic/air-based pressure sensors. While air-based solutions can rely on stronger and repeatable calibrations, they also require a certain space to be positioned between the human limb and the device. Their positioning results in less transparency and could strongly affect the perceived comfort of the user at the test execution. FSR introduces the advantages of a more adjustable and flexible shape and size, together with a reduced price. We consider FSR solutions to have found more applications in the interaction distribution measurement, but they are less reliable in providing an accurate and repeatable output, although this lack could be filled by air-inflatable solutions. Different solutions were limited. Optical-based solutions were fibre-optic sensors and optoelectronic laser pads, but their development was limited to two different authors. Among the remaining technologies, 3D-capacitive sensors could represent a valuable technology for future research given their adaptability and the advantages of the capacitive technologies with respect to FSRs in terms of hysteresis and robustness.

### 4.2. Safety Evaluation

The problem of assessing safety in pHEI remains related to the necessity to properly measure and evaluate the force-pressure exchanged, creating a link between the recorded data and the safety of the contact evaluated. Typically, no specific metrics are described when normal interaction is used for control purposes or for custom sensor validation. Instead, specific metrics are usually found in studies focusing on safety, although information of a singular force component is normally insufficient to address safety. Safety cannot be properly addressed by force and torque metrics, since they do not normally include information on the contact area. The contact safety is evaluable when both the force and contact area are known; thus, pressure metrics need to be included when safety considerations are performed [75].

Accepted metrics in the literature linking pressure with pain and discomfort are: (i) the pressure magnitude at which the pain occurs (pain detection threshold, PDT), and (ii) the pressure magnitude that causes unbearable pain (pain tolerance threshold, PTT) [76]. However, these metrics suffer several influencing factors such as skin condition, age, gender, stimulus, and body site that concur in making the finding of clear pass/fail criteria very challenging.

Typical tests for PDT/PTT definition concern normal loads and static contacts for medical purposes. The subjects are stimulated either in a single point (single point algometry) or on a surface (computerized pressure algometry) [76,77,78,79,80,81]. Due to this evidence, most of the results in this review concerning safety verification considered normal loads and referrred to the mentioned tests in the literature [18,20,25,45,55,56,59,66,68,69].

However, both normal and tangential force components contribute to contact safety [73]. Shear stresses are thought to act in conjunction with normal pressure to produce the damage. From the literature, we know how in the presence of high shear forces, half the normal force normally required for blood vessel occlusion was enough to produce the same effects [82,83,84,85,86]. Shear stresses not only act to decrease the bearable normal pressure level but also concur in blister generation under repetitive rubbing [87].

Despite the fact that soft tissue damage was consistently present in different reviews of adverse events [88], hazards [89], and risk management [90] for lower limb exoskeletons, research on shear stresses is still very limited. Records considering shear pressures and their limit for human safety were sensitively limited.

A shear stress–time relation was already proposed [91], underlying the dependence of the pressure’s effect with time. The importance of considering exposure time together with interaction magnitude lead to the development of safety tests for physical assistant robots [92,93], later adopted in ISO TR 24482-1 [75]. Perceived pressure was also suggested to increase with time, providing different results for ergonomic evaluations [68].

Contrary to this trend in the literature, the collected records compared their results with pressure limits from single point algometry methods, which are appropriate for concentrated rather than distributed loads. FSR arrays were used for pressure contact measurements [25,55], claiming how interaction remained much lower than PTT in the literature [25,55,59,68] However, inconsistencies between perceived pain and PTT threshold were also experienced [55,68]. Furthermore, the use of PTT seems improper in exoskeletons since it is related with unbearable pain, whereas exoskeletons are devices meant to be used for a prolonged period of time.

PDT/PTT-targeted for exosuits were collected through a visual analog scale (VAS) and questionnaires from the participants to inform design specifics [94]. PDT in calves, thighs, and knees spanned from 21.4 kPa to 90.3 kPa, while PTT in the same locations was assessed from 49.6 kPa to 90.3 kPa. In the context of safety, the use of pressure rather than force thus appears more justified since all the available discomfort and pain limits are presented in a pressure scale. PDT and PTT limits overlapped considerably; thus, the effectiveness of the aforementioned pressure ranges will be better explored in real exoskeleton tasks and supported by further evidence.

Variability issues of pressure limits were also experienced at the strapping pressure, and were rarely properly considered. Initial strapping pressure was adjusted at a level considered comfortable for the subjects. However, although both studies considered upper limb exoskeletons, maximum strapping pressure was set at 14 kPa for [18], while the referred ideal pressure was set at 2.6 kPa in [45].

These inconsistencies suggest that different devices might need different strapping forces to guarantee user comfort during the task. If we look at lower limbs, 133 kPa [25] was considered an acceptable pressure, thus suggesting that different interaction pressure studies will be performed for different device families. One more consideration is needed when patients are included in the task. One of the main scopes of an exoskeleton is to assist patients, and physical limits cannot be generalized considering the literature on healthy subjects. From this point of view, only two results in this review included spinal cord injury, SCI patients [25,65], and only one focused on pHEI measurements for safety interaction assessment [25], thereby showing how pressure recorded in SCI patients was greater than healthy patients.

Subject, device, and task seem to highly influence the interaction output, preventing the finding of more generalized safety limits. Still, few studies have been conducted in the field of exoskeletons, leaving a research gap that will be addressed together with exoskeleton development.

## 5. Conclusions

This review summarized the most relevant publications focusing on the assessment of physical human–exoskeleton interaction (pHEI) in the last 10 years.

Apart from the increasing interest of the community in this topic, we identified a clear gap in the definition of protocols and procedures to assess pHEI.

Proposed methods suffer a great variability of tests, protocols, subjects, and setup conditions, making their relationship with human safety limits unclear. The proper, objective, and reproducible study of pHEI could represent a crucial step forward in the field of safety evaluation in wearable robots. Here, studying how the interaction is distributed at the contact points between the human and the robot should be a matter of additional attention from the community. Future studies should consider improving test reproducibility and setup flexibility to cover a wider range of devices and allow harmonization and test comparisons. More solid knowledge of the effect and characterization of interaction pressure is now needed to build protocols that can be applied not only to the single device in a study but to a device family. Additional future efforts should also clarify the effectiveness of kinematic metrics to be used as safety-related indicators.

This review can help researchers understand the current challenges in the assessment of exoskeleton safety and promote cooperative work within the community to find agreed-upon testing methodologies and metrics to properly assess the quality of the existing physical interaction between humans and exoskeletons.

## Figures and Tables

**Figure 1 sensors-22-03993-f001:**
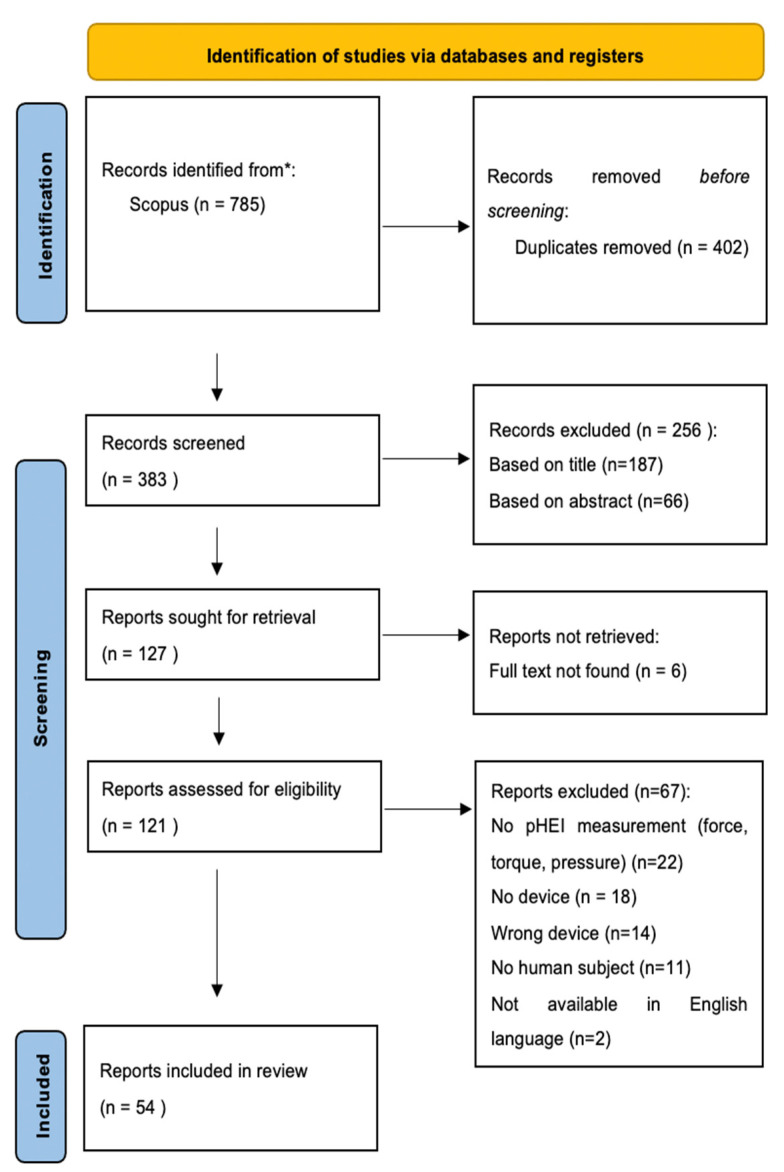
Prisma diagram of the conducted review.

**Figure 2 sensors-22-03993-f002:**
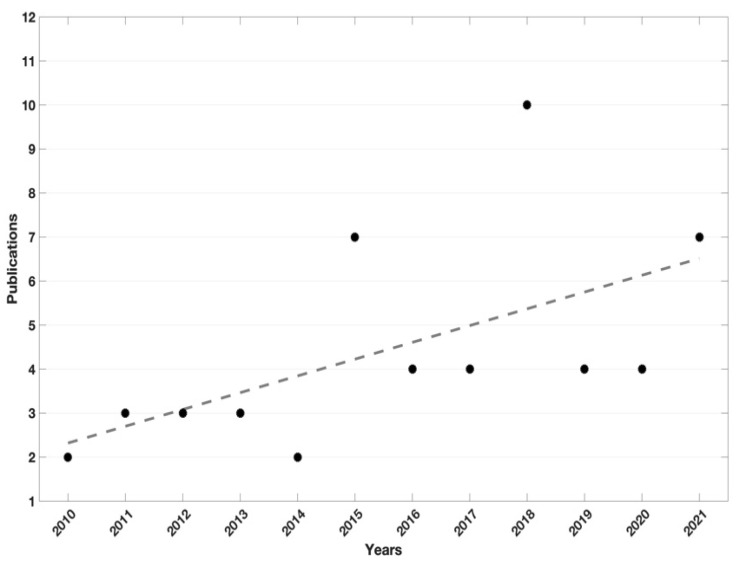
Black dots represents the number of publication per year, dotted line is the black dots trend.

**Figure 3 sensors-22-03993-f003:**
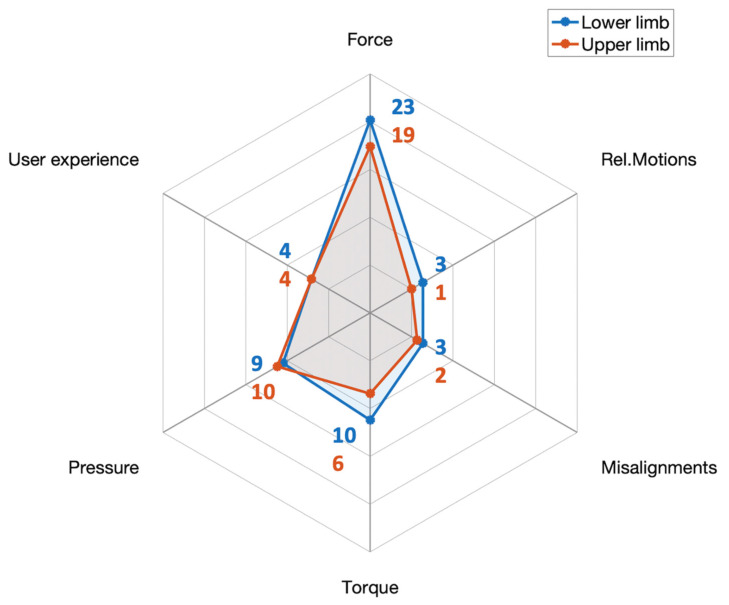
Number of publications including general pHEI metrics divided for upper and lower limb devices.

**Figure 4 sensors-22-03993-f004:**
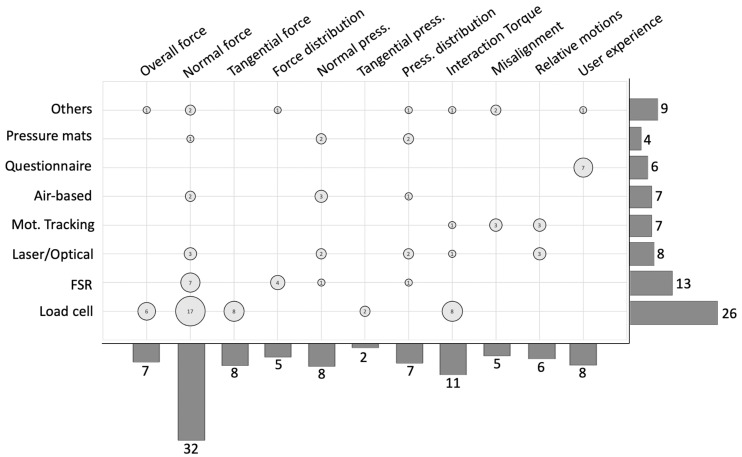
Metrics and sensors solutions in the results. Bar plot on the right represents the number of publications including the listed sensor solutions. Bar plot on the bottom represents the number of publications including the listed metrics. Circles represent the number of studies extracting the relative metric through the relative sensor solution at the intersection.

**Table 2 sensors-22-03993-t002:** Review summary. IF: interaction force, IT: interaction torque, IP: interaction pressure, n.a.: not applicable.

Author and Ref.	Year	pHEI Metrics	Sensor	Protocol	Device	Sensor Placement
Akyiama et al. [46]	2012	IF/support metrics: Normal force, Misalignment	Load cell 3D motion capture system	n.a.	Lower limb exoskeleton frame mounted on a dummy leg	Lower leg Upper leg
Akyiama et al. [64]	2015	IF/support metrics: Normal/Tangential IF, Relative motions	3-axis Load cell 3D optical motion capture system	10 sit-to-stand motions	Leg type motor-actuated lower-limb orthosis	Lower leg Upper leg
Akyiama et al. [69]	2012	IF/IT/support metrics: Normal/Tangential IF, Interaction moment, Skin slippage, Relative motions	3-axis Load cell Slip sensor (2D imaging devices) 3D optical motion capture system	15 sit-to-stand motions	Lower limb physical assistant robot	Upper leg
Amigo et al. [48]	2012	IF/support metrics: Normal/Tangential IF, Misalignment	6-axis Load cell Full bridge strain gauges	Forearm flexion-extension	Arm orthoses	Forearm
Awad et al. [70]	2020	IF/support metrics: Disturbing force, Adverse event observation, Patient feedback	Load cell Questionnaire	20 min of overground walking practice, 20 min of treadmill walking practice	Lower limb soft exosuits	Not specified
Beil et al. [40]	2018	IF: Overall 3D IF	3-axis Load cell	13 different motion tasks	Lower limb exoskeleton	Upper leg Lower leg
Bartenbach et al. [47]	2015	IF/support metrics: Overall IF, Misalignments, Perceived discomfort, Physical load	Load cell 3D optical motion capture system Questionnaire	2 min of familiarization and 20 s of test on a treadmill	Lower limb exoskeleton	Lower leg Upper leg
Bessler et al. [30]	2019	IF/IT: Normal/Tangential IF, Force distribution, Interaction torque	FSR sensor 3-axis load cell	Moving forearm along 3 axis	Forearm support	Forearm
Choi et al. [19]	2018	IF: Normal force	FSR sensor	Treadmill walking	Hip exoskeleton	Thigh
Christensen et al. [43]	2018	IF: Normal force	FSR sensor	n.a.	3DOF spherical mechanism for shoulder joint exo	Arm Forearm
Del-ama et al. [39]	2011	IF/IT: Mean interaction force Mean interaction torque (calculated)	Gauge bridge	10 min leg swing	Lower limb exoskeleton	Lower leg
De Rossi, Lenzi et al. [21]	2010	IP: Pressure distribution	Matrix of optoelectronic sensors	treadmill walk at 4 Km/h in 3 different conditions: “no-assistance” “low-assistance” and “high-assistance”	Lower limb robotic platform	Upper leg Lower leg
Donati, De rossi et al. [9]	2013	IF/IP: Average IF, Maximum IF, Maximum IP	Load cell Matrix of optoelectronic sensors	Upper limb: Passive arm Active arc Lower limb: Transparent mode Viscous field	Elbow active orthoses Lower limb robotic platform	Forearm Upper leg Lower leg
Fan et al. [58]	2013	IF: Max normal IF, Normal IF	Airbags sensor	knee extension to 30° and 60°	Lower limb exoskeleton	Calf
Georgarakis et al. [61]	2018	IF/IP: Normal/tangential IF range, Max normal/tangential IF, Shear IP range	3-axis Force sensor	relax or contract the forearm muscles by grasping a handle according to different force pattern	Upper limb exoskeleton	Forearm
Ghonasgi et al. [35]	2021	IF: Force distribution	FSR sensor matrix	Elbow extensions	Upper limb exoskeleton	Upper arm
Grosu et al. [22]	2017	IF: Normal force	3-axis Force sensor	n.a.	Lower limb exoskeleton	Hip
Hasegawa et al. [23]	2011	IP: Pressure distribution	Active air mat	Arm suspended Arm moving Arm lifting a weight	Upper limb exoskeleton	Forearm
Huang et al. [38]	2015	IF: Total normal force over 4 point	FSR sensor	n.a.	Upper limb power-assist robotic exoskeleton	Forearm
Huysamen et al. [68]	2018	IP/support metrics: Maximum pressure, Local perceived pressure, Subjective usability	Pressure mat Questionnaires	Lifting a load from the ground, with/without device, with/without load	Back powered exoskeleton	Shoulder Hip/lower back Thigh
Islam et al. [33]	2019	IF: Normal force variation	FSR sensor band	Arm liftingh with different payloads	Passive arm exoskeleton	Upper arm
Ito et al. [24]	2018	IF: Force distribution	Tactile sensor	n.a.	Wearable robot for upper limb	Upper arm
Kim et al. [57]	2013	IF: Average normal IF	Load cell	Walking on a mat	Prototype lower limb exoskeleton	Shank
Kim et al. [31]	2021	IP/support metrics: Maximum pressure, Average normalized IP, Misalignment	Air-bladder pressure sensor 3D optical motion capture system	Knee flexion-extension using a pulling cable attached to the foot	Lower limb exoskeleton	Shank
Langlois et al. [18]	2020	IF/IP/support metrics: Normal IF, Strapping pressure, Relative motions, Energy dissipation, Perceived comfort	Air cushion 3D optical motion capture system Visual analog scale	randomly chosen motions at 5 different inflation pressure	7 DOF robotic manipulator	Arm
Langlois et al. [32]	2021	IP: Pressure distribution	3D printed capacitive sensor pads	Lifting weights with arm straight	Upper arm interface	Arm
Leal-junior et al. [26]	2018	IT: Lifting torque	Optical fiber sensor Potentiometer	Free knee flexion and extension	Lower limb exoskeleton	Shank
Leal-junior et al. [27]	2018	IF: Normal force	Optical fiber sensor (Bragg)	Free knee flexion and extension	Lower limb exoskeleton	Shank
Leal-junior et al. [36]	2019	IF: Normal force	Optical fiber sensor Load cell	Free knee flexion and extension	Lower limb exoskeleton	Calf
Lee et al. [41]	2014	IF: Normal/Tangential IF	Load cell	Arm lifting at different load conditions	Upper limb exoskeleton	Handle
Lenzi et al. [20]	2011	IP: Normal IP, Pression distribution	Matrix of optoelectronic sensors	1. leaving arm passive; 2. moving faster (higher frequency) than the robot; 3. moving slower than the robot; 4. imposing higher flexion angle than the robot; 5. imposing a higher extension angle than the robot.	Elbow active orthoses	Forearm
Levesque et al. [42]	2017	IF: Force distribution	FSR matrix sensor FSR sensor	legged deep squats, lunges, as well as stair climb and descent	Lower limb exoskeleton	Thigh Knee Tibia
Li et al. [44]	2019	IF/IT: 3-D IF, Normalized IF over 3-axis, 3-D IT, Normalized IT over 3-axis	6-axis Load cell	Walking on treadmill	Prototype lower limb exoskeleton	Upper limb Lower limb
Lobo-prat et al. [54]	2016	IF/support metrics: Average normal IF, Comfort	Load cell EMG Questionnaire	Elbow flexion-extension movements against gravity	Passive upper limb support	Handle
Long et al. [34]	2017	IT: Interaction torque	Elastic band	Leg swings in the air	Lower limb exoskeleton	Thigh and calf
Long et al. [49]	2017	IP: Contact pressure	Pneumatic gas-bag	40 m walk with (1) passive exo, (2) active exo without gravity compensation, (3) active exo with gravity compensation	Lower limb exoskeleton	Upper leg Lower leg
Long et al. [63]	2018	IT: Interaction torque	Torque sensor	Natural speed of about 0.8 m/s and the maximum velocity up to 4 km/h with 30 kg loads	Lower limb exoskeleton	Knee
Mahdavian et al. [37]	2015	IF: Normal IF	Strain gauges	n.a.	Prototype upper limb exoskeleton	Elbow
Masud et al. [50]	2021	IF/(IT) Module magnitude of normal IF, Calculated IT	6-axis Load cell	n.a.	Arm exoskeleton	Lower arm
Muozo et al. [65]	2020	IT: Bending torque	Load cell 3D optical motion capture system	Normal walking with locked Orthotic knee and actuated Orthotic knee	Leg orthoses	Knee
Quinlivan et al. [66]	2015	IP: Pressure distribution	Pressure mat	n.a.	Soft exosuit	Thigh Hip Belly
Rathore et al. [59]	2016	IF: Maximum normal IF	FSR sensor	two steps forward (a full gait cycle)	Lower limb exoskeleton	Thigh braces Leg braces
Schiele et al. [45]	2010	IF/IT/IP/support metrics: Mean absolute normal IF, Mean absolute IT, Fixation cuff IP, Tracking error, Subjective comfort and workload, Misalignments	6-axis Load cell Pressure interface Questionnaire	visually track a random target on a screen	Upper limb exoskeleton	Forearm
Tamez-Duque J. et al. [25]	2015	IP: Normal pressure, Pressure distribution	FSR pressure pad	sit to stand, walk forward, turn 180° to the right, turn 180° to the left, stand to sit	Lower limb exoskeleton	Upper leg Lower leg
Tran et al. [62]	2014	IT: Interaction torque	Torque sensor Inclinometer	n.a.	Lower limb exoskeleton	Knee
Tran et al. [51]	2021	IF/IT: Normal IF, Performance index: normalized square sum of the sagittal plane IF	2-axis Force sensor	n.a.	Lower limb exoskeleton	Thigh Shank
Wan et al. [56]	2020	IF/IP/support metrics: Average normalized IF (normal + shear), Maximum shear stress, Human-cuff relative motion, Cuff slip velocity	3-axis Force sensor Laser mouse sensor	Walking on treadmill	Custom-made lower limb exoskeleton	Thigh Calf
Wang et al. [55]	2020	IF/IP/support metrics: Average normal force, Maximum normal pressure, Comfort	FSR sensor Questionnaire	10 repetitions of sit-to-stand, standing for 10 min, walking for 10 m, and stand-to-sit	Lower limb exoskeleton	Shin Hands
Wang et al. [28]	2021	IF: Overall IF	Soft pneumatic force sensor	Walking on treadmill	Hip exoskeleton	Thigh
Wilcox et al. [60]	2016	IF: Average peak force	FSR sensor EMG	Two steps forward Two steps backward Two sidesteps	Lower limb exoskeleton	Thigh Lower leg
Wilkening et al. [29]	2016	IF/IT/IP Normal IF Interaction torque Normal pressure	Pneumatic pad 6-axis Load cell	n.a	Forearm interface	Forearm
Xiloyannis et al. [67]	2018	IF/IT/IP Normal IF Interaction torque, Pressure distribution, Pressure peak	Pressure pad	Three flexion/extension movements between 0° and 90°	Elbow exosuit	Elbow
Yousaf et al. [52]	2021	IF: Normal distribution, Average RMS normal distribution	6-axis Load cell FSRs	n.a.	Upper arm exoskeleton interface	Arm
Zanotto et al. [53]	2015	IF/IT: Average normal IF, RMS normal IF, Average IT	6-axis Load cell Potentiometric goniometer	Treadmill walking in inertia, velocity, and alignment conditions	Treadmill-based exoskeleton	Thigh Shank

## Data Availability

Not applicable.

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
