# Peer review of "Characterization and Evaluation of Human–Exoskeleton Interaction Dynamics: A Review"

_sensors, 2022, doi:10.3390/s22113993_

Round 1

Reviewer 1 Report

The review paper is very interesting. 

The comments are as follows:

  1. This paper focused on extracting information related to forces, including torque and pressure in physical human-exoskeleton interactions. However, it is not explicitly clear from the title of the paper.
  2. The pneumatic-base sensor and pneumatic in Fig.4 seems to me not adequate. The word pressure or force should be included in the word. Pneumatic pressure sensor or air pressure sensor etc. seems to me general term.
  3.  In Table2 only the measurement position and method are summarized. Quantitative measurement results of force, pressure or torque are not explained. We can only find limitation of the pressure in Table 3. It is recommended to show us quantitative measurement results of the values. 

Author Response

Dear reviewer

Many thanks for your feedback and the precious comments you provided us. The actions taken are here listed with the comments received:

R: This paper focused on extracting information related to forces, including torque and pressure in physical human-exoskeleton interactions. However, it is not explicitly clear from the title of the paper.

Thank you for this comment. I see the point that physical human-exoskeleton interaction can maybe refer to a broader spectrum of measurements while this work is mainly centered on forces, torques, and pressures. We slightly changed the title: “Characterization and Evaluation of Human-Exoskeleton Interaction dynamics: A Review” hoping to more clearly refer to the dynamics of pHEI.

R: The pneumatic-base sensor and pneumatic in Fig.4 seem to me not adequate. The word pressure or force should be included in the word. Pneumatic pressure sensor or air pressure sensor etc. seems to me general term.

Thank you for this suggestion. I also agree that pneumatic sensors can cover a broad spectrum of devices. I proceeded in to replace the wording with “Air-based pressure sensors”, I think in this sense we can clarify that the sensor  

R: In Table2 only the measurement position and method are summarized. Quantitative measurement results of force, pressure or torque are not explained. We can only find limitation of the pressure in Table 3. It is recommended to show us quantitative measurement results of the values.

Table 3 was part of the discussion and came from one single publication while Table 2 is a comprehensive resume of all the publications included in this review. Table 3 was included to highlight quantitive pressure measurement potentially useful for comparing different studies and to inform the user safety in the task (medical literature about pain onset is indeed centered on pressure thresholds that could onset safety issue contacts). This is not true in the case of forces and torques that hiding the information on the contact area hampers any comparison or safety considerations. However, we understand that having one table in the discussion section might confuse the readers. I proceeded in deleting this table and briefly resume its content in the text.

Hoping these actions could fulfill your precious revision and the explications clarify our choices, we remain open to any further comments and any discussion that could improve our work.

In attachments the revised manuscript

Many thanks

Stefano Massardi

Reviewer 2 Report

Interesting A Review on the exoskeleton. There has been no such article before.

I have a few comments:

Introduction:

Please add the information that the exoskeleton can be used during rehabilitation.

Please add the information that exoskeleton may help to increase the range of motion of individual joints and reduce pain in the elderly and after treatment of musculoskeletal injuries.

Materials and Methods

Please explain why the publications specifically for the period 2000-2021 were selected.

Author Response

Dear reviewer

Many thanks for your feedback and the precious comments you provided us. The actions taken are here listed with the comments received:

Introduction:

Please add the information that the exoskeleton can be used during rehabilitation.

Please add the information that exoskeleton may help to increase the range of motion of individual joints and reduce pain in the elderly and after treatment of musculoskeletal injuries.

Thanks for highlighting these points. I agree that the introduction was lacking some statements about the role of exoskeletons in rehabilitation and their benefit effects on the users. I proceeded adding some more references to at least mention this important aspect in exoskeletons.

Materials and Methods

Please explain why the publications specifically for the period 2000-2021 were selected.

Many thanks also for this consideration. This point deserves a motivation from our side. As I added in the manuscript, considering the recent and fast exoskeleton’s growth of the last year we decided to not include studies older than 10+ years as they are likely considering deprecated and early stage devices that are now better equipped and developed. Similar growth is experienced in the world of sensor technologies.

Hoping these actions could fulfill your precious revision and the explications clarify our choices, we remain open to any further comments and any discussion that could improve our work.

In attachments the revised manuscript

Many thanks

Stefano Massardi
